# Modeling the Complete Dynamics of the SARS-CoV-2 Pandemic of Germany and Its Federal States Using Multiple Levels of Data

**DOI:** 10.3390/v17070981

**Published:** 2025-07-14

**Authors:** Yuri Kheifetz, Holger Kirsten, Andreas Schuppert, Markus Scholz

**Affiliations:** 1Institute for Medical Informatics, Statistics and Epidemiology, University of Leipzig, Haertelstrasse 16-18, 04107 Leipzig, Germany; holger.kirsten@imise.uni-leipzig.de; 2Center for Scalable Data Analytics and Artificial Intelligence (ScaDS.AI) Dresden/Leipzig, University of Leipzig, Humboldtstraße 25, 04105 Leipzig, Germany; aschuppert@ukaachen.de; 3Institute for Computational Biomedicine II, MTZ building, Pauwelsstraße 19, University Hospital RWTH Aachen, 52074 Aachen, Germany; 4Faculty of Mathematics and Computer Science, University of Leipzig, 04109 Leipzig, Germany

**Keywords:** COVID-19, SARS-CoV-2 epidemiologic models, dark figure, parameter heterogeneity, parametrization, extended multi-compartment SIR-type model, input–output non-linear dynamical system, Bayesian knowledge synthesis, machine learning, pandemic preparedness

## Abstract

**Background/Objectives**: Epidemiological modeling is a vital tool for managing pandemics, including SARS-CoV-2. Advances in the understanding of epidemiological dynamics and access to new data sources necessitate ongoing adjustments to modeling techniques. In this study, we present a significantly expanded and updated version of our previous SARS-CoV-2 model formulated as input–output non-linear dynamical systems (IO-NLDS). **Methods**: This updated framework incorporates age-dependent contact patterns, immune waning, and new data sources, including seropositivity studies, hospital dynamics, variant trends, the effects of non-pharmaceutical interventions, and the dynamics of vaccination campaigns. **Results**: We analyze the dynamics of various datasets spanning the entire pandemic in Germany and its 16 federal states using this model. This analysis enables us to explore the regional heterogeneity of model parameters across Germany for the first time. We enhance our estimation methodology by introducing constraints on parameter variation among federal states to achieve this. This enables us to reliably estimate thousands of parameters based on hundreds of thousands of data points. **Conclusions**: Our approach is adaptable to other epidemic scenarios and even different domains, contributing to broader pandemic preparedness efforts.

## 1. Introduction

During the last years, the SARS-CoV-2 pandemic imposed a worldwide high disease burden, and it is still a relevant contributor to severe infectious lung diseases at a population level. Understanding SARS-CoV-2-induced dynamics at several levels of data is of high importance for proper risk management, including planning of vaccination campaigns, non-pharmaceutical countermeasures, clinical resources, and general pandemic preparedness. A plethora of biomathematical models were proposed for that purpose [1,2,3,4,5,6,7]. Most of these models only describe the pandemics for a limited set of data or a limited time frame [8]. Recently, Burch et al. reviewed mathematical models of COVID-19 vaccination in high-income countries and specifically discussed 47 works [9]. Most of them used deterministic, compartmental models for European countries or North America with a simulated time horizon of 3.5 years or less. Common outcomes included infection numbers, hospital burden, and COVID-19-related deaths. Two of the models describe the pandemic situation in Germany [10,11]. Koslow et al. [11] analyzed the relaxation of non-pharmaceutical interventions (NPIs) under the vaccination campaign in Germany for the period between June 2021 and March 2022. For the spread of SARS-CoV-2, the authors employed an SIR-type model. Rodiah et al. [10] developed an age-structured deterministic SEIRS model to understand the age- and setting-specific contribution of contacts to transmission during the first 1.5 years of the COVID-19 pandemic in Germany. A pre-pandemic contact matrix has been used there. The model was optimized to fit age-specific SARS-CoV-2 incidences reported by the German National Public Health Institute (Robert Koch Institute).

We recently proposed a universal approach to parametrize mechanistic epidemiologic models using multiple, often biased epidemiological or clinical data sets. This approach is based on embedding an epidemiologic model as a hidden layer into an input–output non-linear dynamical system (IO-NLDS, [12]), where the input layer represents factors not known by the model, such as changing non-pharmaceutical interventions (NPIs), vaccination campaigns, the occurrence of new variants, lags in reporting of cases/events [13], and changing testing policies [14]. The output layer represents different types of observational data, which are linked to the hidden layer via so-called data models addressing uncertainty and bias of the observational data in relation to modeled state parameters. Unknown model parameters can be estimated by a Bayesian approach using prior information of parameter ranges derived from different external studies and other available data resources.

By this approach, we were able to describe pandemic dynamics of Germany until April 2021 using an age-structured SIR-type model as a hidden layer. Several changes in the pandemic situation required an update of the underlying epidemiologic model. These changes comprise, for example, (1) the different contact behavior of age groups; (2) modeling of final disease states of age groups; (3) new replacement dynamics of variants, including the possibility of more than two highly prevalent variants at the same time; and (4) modeling of age-dependent vaccination efficacy and, most importantly, immunity waning.

Moreover, several new data resources became available or were improved during the pandemic, requiring new or updated data models to be linked with the updated epidemiologic model. We integrated, for example, data on the progression of the vaccination and booster campaigns per age group and considered respective differences between German federal states. We collected and included external study data on vaccination efficacy, waning dynamics, and booster efficacies, as well as seropositivity studies. We also considered the heterogeneity of pandemic dynamics across states, allowing for the estimation of state-wise variability of epidemiologic model parameters for the first time.

Finally, we improved our modeling architecture in order to speed up and parallelize data processing and calculations, coping with the high-dimensional data and parameter space, facilitating our proposed full information approach.

## 2. Materials and Methods

### 2.1. General Approach

We consider an input–output non-linear dynamical systems (IO-NLDS) originally proposed as time-discrete alternatives to pharmacokinetic and dynamic differential equations models [12,15]. This class of models couples a set of dynamical input parameters, such as external influences and factors, with a set of output parameters, i.e., observations by a hidden model structure to be learned (named *core model* in the following). This coupling represents a hybrid modeling concept, in which deterministic model equations reflecting our mechanistic understanding of the pandemic are combined with empirical relationships of state variables and observational data called *data models* in the following. This represents a major feature of our approach because it allows for the separation of the tasks of epidemiologic model development and addressing of data issues, such as corrections for incomplete or biased observational data prior to model parametrization.

### 2.2. Concepts and Assumptions of the Core Model

Our core model is of SECIR type (SECIR = susceptible, exposed, hospitalized COVID-19 cases, infectious, recovered subjects) [16] and consists of several sub-models to account for heterogeneity of model parameters and to resemble infection histories. These sub-models are characterized by up to three different attributes, namely age group; immune status (*is*), including its waning after vaccination or infection events; and virus variant (*vv*).

More precisely, we considered the following categories of these features:Five age groups: 1–14 years, 15–34 years, 35–59 years, 60–79 years, and ≥80 years;Ten virus variants: WT (wild type); alpha; delta; and omicron BA1, BA2, BA5, and BA.2.75 with BQ.1, XBB, BA.2.86, and KP.3;Four immune statuses: Naïve due to absence of vaccinations or infections (*S*) or shortly after first vaccination (*Vac*_0_), highly protected by either recent vaccination (*Vac*_1_) or recovery from a recent infection (*R*_1_), moderately protected (*Vac*_2_, *R*_2_), and weakly protected (*Vac*_3_, *R*_3_), see Table 1.

We assume four vaccination statuses to account for the fact that protection against infection is incomplete and wanes much faster than protection against critical disease courses (see Appendix A). Assignment of attributes to compartments is modeled as multidimensional parameter arrays (tensors).

We distinguish three major compartment groups by their assigned attributes: (1) infectible subjects without previous infection events (*Sc*, *Vac*) and attribute age; (2) infectible subjects with previous infection events (*R*) and attribute age with a previous virus variant (one of the ten variant (groups): WT; alpha, delta BA1, BA2, BA5, BA.2.75/BQ.1, XBB, BA.2.86, KP.3) or immune status prior to previous infection (naïve, highly, moderately, or weakly protected); and (3) infected subjects (*E*, *I*) with attribute age, virus variant, and immune status prior to infection. For the latter, we assume that only *I* is contagious. Compartments *H* and *C*, representing patients admitted to hospital ward or intensive care unit (ICU), are only counted compartments. They have the same attributes of their originating compartment *I* and represent a second hidden layer of our model, which is later connected to respective observational data.

In more detail, we make the following assumptions.

Infected compartments are those carrying a specific virus variant. This applies to the compartments *E*, *I*, *H*, and *C*;The latent state *E* comprises infected but non-contagious subjects. This is the transient state between becoming infected and becoming contagious;To model time delays in transitions, we frequently divide compartments into sub-compartments with first-order transitions. This approach is extensively used in pharmacological models [17]. It was shown that this approach resembles Gamma-distributed transit times [18];The infected state *I* is the only state assumed to be contagious and is divided into four sequential compartments. There is a single branching for the compartment *I*_2_, from which patients can proceed either to *D* (death compartment, representing deaths due to COVID-19) or to *I*_3_. Finally, the efflux of *I*_4_ enters *R*_1_, representing resolved disease courses;All sub-compartments of *I* contribute to new infections, depending on age, virus variant, and immune status of target subjects;The compartment *I*_2_ is considered the source of severe disease outcomes, comprising treatment at hospital wards *H* or ICU (*C*). These contributions are not modeled by fluxes but as counting respective bed occupancies;The compartment *H* represents disease states requiring hospital ward care. We assume that these patients are not infectious due to isolation. The compartment is divided into three sub-compartments, *H*_1_, *H*_2_, and *H*_3_, to allow comparisons with data on hospital ward bed occupancies. *Rhosp* counts resolved disease courses after hospital ward station care;The compartment *C* represents critical disease states requiring intensive care. Again, we assume that these patients are not infectious due to isolation. In analogy to the compartment of hospital ward treatment, this compartment is also divided into three sub-compartments to mimic disease courses, allowing for a comparison of the compartment with data of ICU bed occupancies. *Ricu* counts resolved disease courses after critical state to model cumulative data.

Basic qualitative properties of infectible compartments assigned with different immune statuses are provided in Table 1. Respective transitions are displayed in Figure 1. More details of model compartments and their properties are provided in Appendix A.

All model assumptions are translated into a difference equation system (see Appendix A). The mathematical structure is that of an input–output non-linear system as depicted in Figure 1. Relationships between immune statuses obtained by vaccination or a previous infection event and possible disease courses are displayed in Figure 2. Model parameters are explained in the Appendix A and are listed in the respective tables: (1) basic epidemiologic parameters (Appendix A), (2) disease severity parameters (Appendix A), (3) virus variant modeling (Appendix A), (4) contact matrices (Appendix A), and (5) input layer parameters (Appendix A).

#### 2.2.1. Input Layer

Here, we describe the structure of the input layer of our IO-NLDS. This layer is designed to model the impact of external factors acting on the epidemiologic dynamics, such as changing infection rates due to non-pharmaceutical interventions, vaccination and booster campaigns, and changing testing policies. Effectively, these input functions dynamically affect parameters of the hidden layer containing the epidemiologic model. We describe these different external factors in the following. Respective parameters are described in Appendix A–S6,S7,S9.

*Dynamical infection rate:* We define a step function b_1_ as time-dependent input parameter modifying the rate of infections. To identify time points of steps, we used a data-driven approach based on Bayesian Information Criterion (BIC) informed by time points of governmental changes in non-pharmaceutical countermeasures in Germany, changing testing policies, as well as events with significant impact on epidemiological dynamics, such as holidays or sudden outbreaks [12]. Details can be found in Appendix A.

Daily testing and number of undetected cases (estimation of dark figure): It is well-known that reported numbers of infections are largely underestimated and that this bias is time-dependent during the pandemic. We estimate the dark figure (DF) based on calibration analyses of seropositivity data and near-representative systematic testing from the SentiSurv study (see Appendix A for details). Respective time-dependent estimates are used to cast true infection numbers and compare them with respective epidemiologic model compartments.

*Vaccination and booster campaigns:* Numbers of applied vaccination and booster doses are available from the German Robert Koch Institute on a daily scale and per age group and federal state. We distribute respective vaccination rates over eligible model compartments according to their relative size.

#### 2.2.2. Output Layer, Data, and Parameter Fitting

Unknown parameters of the model are determined by parameter fitting. For this purpose, compartments of our hidden layer epidemiologic model are coupled with observational data via the output layer of our IO-NLDS using appropriate (stochastic) link functions called *data models*. We present these data, respective data models, and objective functions in the following.

We fit our model to age- and federal state-specific time series data of reported numbers of infections *I_M_*, occupation of hospital stations *N_M_*, occupation of ICU beds *C_M_*, and deaths *D_M_*, representing the output layer of our IO-NLDS model. Moreover, we fitted data of the variant dynamics.

Data sources of infections, normal ward admissions, and deaths were publicly available from the Robert Koch-Institute (RKI). For our modeling, we used data from 4 March 2020 to 12 September 2024. Number of critical cases, i.e., ICU admissions, was retrieved from the German Interdisciplinary Association of Intensive and Emergency Medicine (Deutsche Interdisziplinäre Vereinigung für Intensiv- und Notfallmedizin e.V.—DIVI) for the time window of 25 March 2020 to 12 September 2024. Time points in proximity to Christmas and the turn of the year 2020/21 (i.e., 19 December 2020 to 19 January 2021) were heavily biased and therefore discarded during parameter fitting.

Despite this fact, data are considered to be largely biased, i.e., they cannot be directly linked to state parameters of our epidemiologic model. This was addressed by the following pre-processing steps and data models, aiming at removing major sources of bias.

Infected cases: We first smoothed reported numbers of infections with a sliding window approach of seven days centered at the time point of interest to remove the strong weekly periodicity of the data. We assume that these registered numbers correspond to a certain percentage of infected patients. To project true infection numbers, we estimate the time-dependent dark figure as explained above. We further account for delays in the reporting of case numbers by introducing a log-normally distributed delay time, as explained in Appendix A.

*Deaths and hospital ward admissions:* Deaths and hospital ward admissions were reported at a daily scale by the RKI since the beginning and end of March 2020 [19], respectively. However, due to data privacy, RKI did not provide exact dates of deaths or hospital ward admissions. Rather than this, reported dates of death and hospitalized patients correspond to the dates of reported infections of these patients. We aimed to remove the resulting reporting delay by assuming log-normally distributed delay times where the expected delay time is derived from the respective transit times of our model. Details can be found in Appendix A.

*Critical cases:* Number of critical COVID-19 cases (DIVI reported ICU) was available at the end of March 2020 [20]. We assumed that these data were complete on 16 April 2020, when reporting became mandatory by law in Germany. Earlier data were extrapolated from the number of reporting hospitals using the total number of ICU beds available according to the reported ICU capacity in 2018. These estimates are coupled with the sum of critical sub-compartments *C_i_* (*i* = 1, 2, 3) of our model.

*Frequency of variants:* We consider ten virus variants or variant groups constituting major waves in Germany, namely WT; alpha; delta; and omicron BA1, BA2, BA4 + 5, BA.2.75/BQ.1, XBB, BA.2.86, and KP.3. Entry of new variants is modeled by an instantaneous influx of infected subjects into the compartments *E* and *I* [21]. Respective parameters are estimated separately for each federal state. For the purpose of parameter fitting, we also consider data of variant frequencies available for Germany, taken from public reports of the RKI [22]. Details can be found in Appendix A.

#### 2.2.3. Parametrization Approach

We carefully searched the literature to establish ranges for mechanistic model parameters. We considered two alternatives for parameter estimation: usage of these data as prior information for a Bayesian approach (implemented in our earlier work [12]) or setting the parameters from the literature as fixed values (Appendix A). Comparison of goodness of fits favored the second alternative. Parameter estimation is achieved via likelihood optimization. The likelihood is constructed using similar principles as reported previously [23]. In short, the likelihood consists of three major parts, namely a penalty term to ensure that model parameters are within prescribed ranges, as well as penalties for the variability of parameters across federal states as explained in Appendix A. Penalization of variability of parameters across federal states is some kind of pruning to avoid overfitting [24]. We follow a full-information approach intended to use all data collected during the epidemic, as explained in Appendix A. Consequently, our parametrization approach is intended to describe complete dynamics of the epidemic in Germany and its federal states in the time period covered by the data (Appendix A).

Likelihood optimization is achieved using a variant of the *Hooke–Jeeves* algorithm [25]. This is a zero-order algorithm, which does not require expensive calculations of derivatives of the fitness function to be optimized. In brief, the method relies on iterated updates of current fitness values and respective parameter settings by comparisons with fitness values in the neighborhood of the current parameter settings, separately for all coordinates. Perturbation sizes at each dimension are adapted in dependence on the result of the previous iteration step, i.e., a perturbation in the *s*-th dimension becomes larger if a better fitness value was found for this dimension in the previous iteration. Otherwise, it is reduced in the next step, provided that it does not drop below a specified lower limit. An exception is parameters related to the time of entry of new variants, for which we used a constant step size of one day in order to stabilize convergence, since these parameters are highly sensitive. The algorithm stops if the last four steps did not provide a relative improvement of the fitness function of more than a specified tolerance parameter *δ_tol_*.

Identifiability of parameters was checked using profile likelihood technique [26]. Details are explained in the supplement (Appendix A). Dynamical parameters were determined by step functions. The number of these steps was determined based on pre-described events or empirically. The Bayesian Information Criterion was applied to penalize the number of steps.

#### 2.2.4. Implementation

The model and respective parameter estimation procedures are implemented in the statistical software package R version 4.2.2, from which external publicly available functions are called. The model’s equation solver is implemented as C++ routine and called from R code using the Rcpp package.

## 3. Results

We aim to explain the dynamics of the COVID-19 pandemic regarding infected subjects, hospital ward and ICU occupation, deaths, and variant frequency for the entire time period between 4 March 2020 and 12 September 2024 for Germany and its federal states. We first present the results of our parameter fittings. We then show the resulting agreement of model and data for Germany and its federal states and discuss the dynamics of immune statuses. Based on the observation of largely deviating dynamics between federal states, we analyze respective heterogeneity in data and parameters in more detail. Finally, we present a number of validated model predictions.

### 3.1. Parameter Fitting and Identifiability

Most of the model parameters were obtained by fitting the predictions of the model to available data on infection dynamics, hospital burden, deaths, and variant frequencies. Non-identifiable parameters according to profile likelihood examination were set constant. Overfitting was further controlled by a BIC-based model selection process (see methods).

Likelihood profiling showed that most of the mechanistic parameters of the SECIR model, such as the transition rates $r_{3}$, $r_{5}$, $r_{6}$, and $r_{7}$, could be fixed to the values derived from other studies (Appendix A). Sensitive parameters were $r_{1}$ representing the basic infection rate and $r_{9}$ representing the hospitalization rate.

The results of parameter estimates can be found in Appendix A (hospitalization) and Appendix A (ICU).

### 3.2. Comparison of Model Predictions and Observed Data for Germany and Its Federal States

Throughout the pandemic, we observed a good agreement between the modeled state variables, respectively, their linked output layers, and the data, as shown in Figure 3. In Figure 3A, we present the dynamics of infection numbers for the five considered age groups and overall. Respective dynamics of hospitalized, ICU, and death cases are presented in Figure 3D. The agreement between the model and data was uniform for the different age groups. The model is also in good agreement with the dynamics of virus variant frequencies (Figure 3C), thereby effectively separating the impact of the variant on the transmission dynamics from other reasons of differing infectivities, such as contact behavior or immunity and its waning. Results of the 16 federal states fit similarly well (Appendix A).

We account for temporarily differing unreported cases specific to age groups and federal states by estimating a time-variant dark figure based on the reported percentage of positive tests and seroprevalence data. It revealed that the DF is subject to considerable changes. During periods of intensive testing, it ranged between 50% and 150% but became much larger thereafter (see Appendix A for details).

Temporal changes of infectivity not explained by virus variants, dynamics of the dark figure, and dynamics of immune statuses of subjects are attributed to a stepwise dynamic change of transmission parameter *b*_1_ (Figure 3B panel). This parameter reflects the strength of contact inhibition at different phases of the pandemic due to NPIs or changes in individual contact behavior, e.g., due to holidays or increased awareness. It is specific for age groups and federal states. For more details, see the description of the residual infectivity in Appendix A.

### 3.3. Dynamics of Immune States and Their Impact on Severe Disease Courses

In our model, (re-)infection probability and susceptibility to severe courses of infection do not only depend on the attacking virus variant but also on the immune states of subjects, including their immunization history. Here, we distinguish four different immune states: (1) the immunological naïve state (*S*/*Vac_0_*) with the highest susceptibility to infection and a severe course; (2) high and (3) moderate protection either due to vaccination (*Vac*_1_ and *Vac*_2_, respectively) or previous infection (*R*_1_ and *R*_2_, respectively); and (4) low protection due to immunity waning (*Vac*_3_, *R*_3_). The dynamics of these modeled immune states for Germany are shown in Figure 4 and separately for the federal states in Appendix A. Age group-specific immune states are presented in Appendix A.

We estimate, for example, that the percentage of infected subjects was about 5% by the end of the year 2020, while the percentage of susceptible unvaccinated subjects was <20% by the end of 2021 and dropped to almost zero by the end of 2022. We also estimate that a small percentage of subjects were vaccinated and never infected (*Vac*_3_) by the end of the simulation period. As expected, percentages of most strongly protected subjects (*R*_1_) rise during a wave and decline thereafter.

These dynamics affect the courses of age-specific numbers of hospitalized patients, as shown in Figure 3D for Germany and separated for the federal states in Appendix A.

### 3.4. The SARS-CoV-2 Pandemic in Germany Exhibited Strong Regional Heterogeneity

The heterogeneity of federal states was assessed up to the BA4 + 5 wave, and epidemic monitoring was reduced thereafter. Epidemic dynamics differed considerably between states, which could not be fully explained by differences in vaccination campaigns or age distribution. In this section, we analyze this heterogeneity in more detail by comparing the respective data and model results.

We assumed between-state heterogeneity for parameters related to severity, i.e., rates affecting the hospitalization (*N*), ICU (*C*), and death (*D*) compartments. We also assumed state-specific parameters related to entries of new virus variants, such as timing and initial infection numbers. Moreover, the dynamics of the residual infectivity *b*_1_ were assumed to be state-specific. The selection of state-specific parameters was again based on BIC. Of note, it turned out that no differences in viral properties need to be assumed, but that the majority of state-specific parameters correspond to population structure and pandemic management, which indeed was heterogeneous across states. To avoid overfitting of state-specific parameters, we penalized deviations from the estimates obtained for Germany.

We first analyzed the heterogeneity of the 16 German federal states with respect to infection numbers, deaths, and test positivity (Figure 5A–C). Sub-figures show dynamics of incidences, cumulative deaths, and positive test rates for the different states and Germany as a whole. For example, the federal state of Saxony had a death burden four times higher than that of Schleswig-Holstein. Testing policy varied across regions, as evidenced by differences in test positivity rates of up to 75%. Moreover, DF estimates differ considerably during the pandemic and between federal states.

As examples of parameter heterogeneity, Figure 5D shows state-specific estimates of parameter values for the transfer rates from the compartment of infected subjects *I*_2_ to the death compartment *D*, also depending on age group and virus variants. Estimates of differences in the rates to develop a severe course are shown in Appendix A and appear to be plausible: Exemplarily, federal states Saxony and Thuringia show the highest parameter estimates for transition rates to the compartment *D* (death) in the oldest age group 80+ during the Delta and Omicron-BA1 wave. Correspondingly, both regions are reported to have suffered from high excess mortality in the elderly. Furthermore, death rates of the Omicron variant are consistently lower than those of previous variants, which is again in line with the literature. Summarizing our findings, disease dynamics vary across states primarily because of variations in the timing and magnitude of variant introductions; differences in infectivity, which are likely contact intensity-driven; and differences in severity parameters.

In Figure 6, we compared the between-state heterogeneity of infection dynamics with the residual infectivity estimates. We show dynamics of incidences in Figure 6A and corresponding residual infectivity estimates (Figure 6B) for the 16 federal states and Germany. Phases of strong heterogeneity of incidences are marked and correspond to higher heterogeneity in residual infectivity.

We also estimated the number and timing of introductions of new SARS-CoV-2 virus variants for each federal state and Germany (Figure 7, panels correspond to the virus variants). Introduction times varied between two and four weeks, explaining time shifts in the incidence curves. Moreover, a considerable heterogeneity of initial entries was estimated. These were remarkably high for the alpha variant, reflecting insufficient NPI at the beginning of the pandemic.

### 3.5. Validated Model Predictions

We regularly used our model to predict scenarios of the future course of the epidemic and published these predictions via our website in 24 bulletins (https://www.health-atlas.de/studies/59, accessed on 16 June 2025). Here, we present comparisons of our predictions with the actual course of the pandemic in order to validate our model and to demonstrate its utility.

*Comparison of the lockdowns of December 2020 and November 2021 in Saxony:* We used our model to predict the impact of lockdown measures on residual infectivities of our age groups. Reduction of residual infectivity after the initiation of stronger NPI measures in November 2021 (lockdown) is shown in Figure 8A for each age group. Comparing the lockdowns implemented in December 2020 and November 2021 in Saxony revealed that similar reductions were achieved, except for the youngest age group. Indeed, this is plausible because schools and day care facilities remained open in the November 2021 lockdown compared to the December 2020 lockdown.

Based on these estimates, we predicted the further course of the pandemic after initiating the November 2021 lockdown and compared the results with a scenario without lockdown measures. Respective daily positive tests, cumulative deaths, and ICU bed-occupancy are shown in Figure 8B and were compared with the actual course of the pandemic. We estimated that the lockdown might have saved a four-digit number of lives. In addition, the lockdown might have roughly halved the peak in ICU bed occupancy compared to the scenario without lockdown, assuming that these ICU bed numbers could be supported. Finally, the further course of the pandemic closely resembles our model prediction for several weeks.

*Impact of higher vaccination rate:* In Figure 9, we show the results of another modeled scenario to assess the impact of the higher vaccination rate in the federal state of Saarland compared to the average German vaccination rate. We modeled the time course in Saarland in two scenarios, one with the actual high vaccination rate and one with the reduced vaccination rate of Germany as a whole. In Figure 9A, we show infection dynamics and cumulative deaths, while in Figure 9B, we show the dynamics of vaccinations. In the higher vaccination scenario, infections in the delta wave were significantly lower, while infections in the later omicron BA4 + 5 wave were significantly higher. This is likely due to the immune escape of the omicron variants regarding vaccination. However, the total number of deaths remained lower in the scenario observed with the higher vaccination rate, consistent with the reported lower pathogenicity of the Omicron variant. This suggests that the higher vaccination rate in Saarland may have also played a relevant role in preventing severe outcomes in the long term.

## 4. Discussion

In this paper, we refined and extended our approach of embedding and parametrization of a SECIR-type model to explain the complete course of the SARS-CoV-2 epidemic in Germany as well as in all its 16 federal states [12]. The general idea is to embed differential equation-based epidemic modeling into an input–output dynamical system (IO-NLDs), combining explicit mechanistic models of epidemic spread and phenomenological considerations of external impacts on model parameters via the input layer. We also assumed a non-direct link between state parameters of the embedded SECIR model and observables via data models that effectively address known biases of available data resources.

Here, our proposed model updates a previous one by assuming age-dependent sub-models with a known underlying contact matrix and extension of virus variant-dependent sub-models from two to unlimited (currently ten). We considered the effects of vaccination and immune waning by some kind of immunologic clock, which is set back by immunization events. The resulting updates enable us to remove two of three empirical dynamical parameters required in our previous model, namely *p_crit_* and *p_death_*, representing variations in the probability of developing critical symptoms or dying. This shows how unexplained empirical inputs can be later mechanically modeled when additional information becomes available.

We also included modeling certain health care requirements, in particular, occupancies of the regular ward and ICU. Both were modeled by additional hidden layers of our IO-NLDS serving as count compartments.

Based on our IO-NLDS formulation and data models, we parametrized our model on the basis of data on infection numbers, bed occupancies at regular wards, and ICU deaths available for Germany and its federal states. Here, we chose a full-information approach considering all data from the start of the epidemic on 4 March 2020 to 12 September 2024, significantly increasing the modeling period compared to other available models for Germany [10,11]. We also applied a Bayesian learning process by considering other studies to inform model parameter settings. Thus, we combine mechanistic model assumptions with results from other studies and observational data. This approach is very popular in pharmacology [17], but despite its importance, it is yet rarely applied in epidemiology [18].

Model parametrization resulted in a good and unbiased fit of data for the period considered for Germany and its federal states. A total of 37 intensification and relaxation events were necessary to describe the epidemic dynamics over the time course of observations for each federal state and each age category, i.e., 85 different dynamical parameters of residual infectivity *b*_1_ were required. We estimated higher values of *b*_1_ at the very beginning of the epidemic, which could be due to natural contact behavior, but could also be caused by issues regarding reporting or lack of testing capacities, i.e., we cannot exclude that this is an unresolved data artifact. Estimated infectivity roughly correlated with the Governmental Stringency Index [27].

Seasonal effects were not explicitly considered in our model. Seasonal changes are discernible only in 2020. In 2021, the seasonality could not be clearly detected due to interacting NPIs. In this year, the variability of *b*_1_ between federal states was large, exceeding the variability among age groups for the overall data of Germany. This is plausible because NPI implementation was very heterogeneous between states. Only later in the pandemic (end of 2021) were attempts to implement nationwide harmonization of NPI rules and measures (see Appendix A). In 2022, *b*_1_ was less than 0.2 most of the time in all age categories and relatively stable with little variability between federal states, which might mirror this nationwide harmonization of NPI rules.

We also demonstrated the utility of our model using several mid-term simulations of scenarios of epidemic development in Saxony, a federal state of Germany. We could show that predictions of reported infections were in the range of later observations for scenarios considered likely. Forecasts of our model were regularly contributed to the German forecast hub [28].

Further extensions and improvements of our model are conceivable. One might consider stochastic effects on a daily scale, for example, to model random influxes of cases or to model random extinctions of infection chains. These effects are relevant in times of low incidence numbers, such as those observed in Germany in the summers of 2020 and 2021. Our IO-NLDS framework is well-suited to implement such extensions [12]. Another possible application of our model could be to provide infection scenarios for other types of models, such as hospital models, as successfully demonstrated for German ICU admission forecasts or resource planning tools currently being developed. Finally, so far, we have not considered births, non-COVID-19 related deaths, and aging in our model. This is justified by an almost constant distribution of our age groups during the course of the pandemic (Appendix A) but could be refined in later versions of the model.

In summary, we proposed a holistic epidemiologic modeling framework to explain the complete dynamics of SARS-CoV-2 in Germany and its federal states and to use it for scenario simulations and predictions. Our approach effectively allows for distinguishing between mechanistic modeling and data-related tasks such as bias analysis and linkage of observational data with model state parameters. Effects acting on the dynamics could be explicitly described by the embedded mechanistic model or phenomenologically imposed via the input layer, allowing a large flexibility of models considered. We believe that our approach is useful not only for the parametrization of the SECIR model presented here but also for other epidemiologic models, including other disease contexts and data structures.

## Figures and Tables

**Figure 1 viruses-17-00981-f001:**
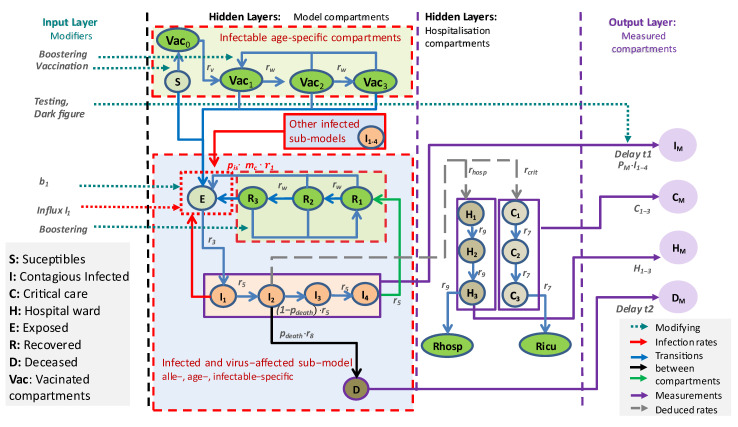
General scheme of our epidemiologic model. Our epidemiologic SECIR model is embedded as a hidden layer into an IO-NLDS. Respective equations are provided in Appendix A. Compartments are grouped according to whether they correspond to infectible (green) or infected (blue) subjects. Attributes of sub-models (i.e., *age*, *vv*, and *is*) are not displayed for simplicity. The infectible compartments without previous infection events (S, *Vac*) only depend on age, while the other compartments depend on age, virus variant, and, if applicable, on the last infection event. The input layer consists of external factors acting on the epidemic, such as vaccination campaigns or parameter changes due to changes in testing policy, and non-pharmaceutical interventions. The output layer is derived from respective hidden layers via stochastic relationships (called *data models*, see later). The output layer is compared with real-world data. The number of hospital (*H*) and ICU (*C*) admissions is described as additional hidden layers, counting these events and describing dynamics of bed occupancies.

**Figure 2 viruses-17-00981-f002:**
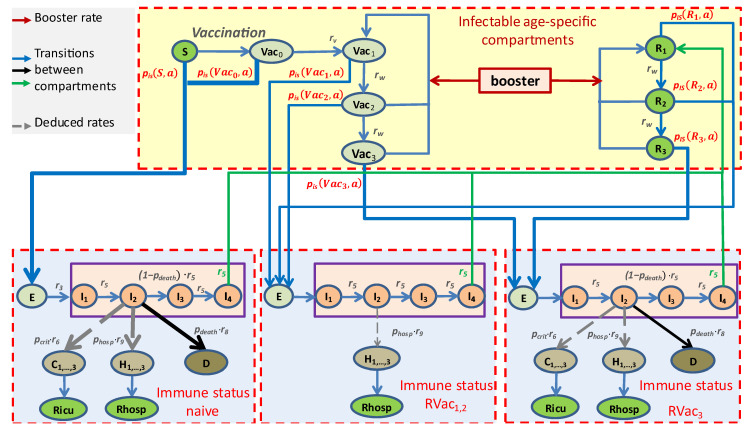
General scheme of the relationships between immune status, infection, and severity of disease course. Probability and course of infection depend on the current immune status. Three immune statuses are distinguished: (1) immune-naïve and freshly vaccinated subjects (left), (2) subjects with recent vaccination or freshly recovered subjects (middle), and (3) subjects with waning immune protection (right). Transition probabilities ***p_IS_***(***Z***,***a***) depend on the current immune status and the infecting variant *a*. Stronger risks are illustrated by broader arrows.

**Figure 3 viruses-17-00981-f003:**
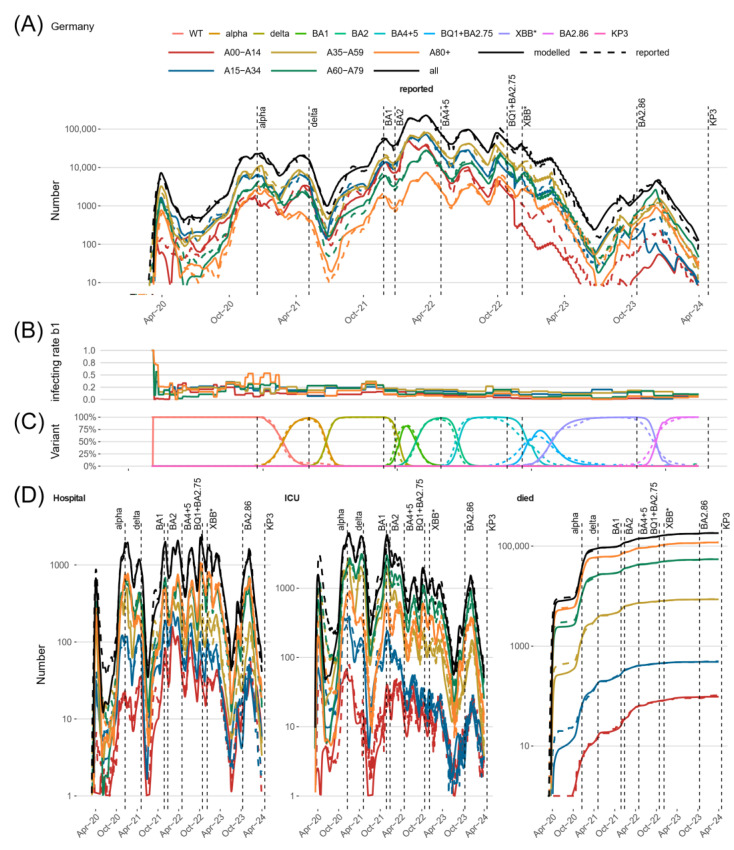
Agreement of model and data for Germany. We present the pandemic phase of Germany between 4 March 2020 and 12 September 2024 and compare observations (dashed lines) with the predictions of our IO-NLDS model (solid lines). We present the overall dynamics and those of the five age groups considered. (**A**) shows good agreement between the model and the reported incidence of positive tests for all age groups and overall. (**B**) shows the dynamical parameter of residual infectivity *b*_1_ for the different age groups. (**C**) shows the agreement between the model and the data of variant frequencies. In (**D**), we present the model/data comparisons for age-specific dynamics of severe disease states, i.e., hospital ward and ICU occupation and cumulative deaths. XBB*: XBB including subvariants. Respective figures of the federal states are provided in the supplement.

**Figure 4 viruses-17-00981-f004:**
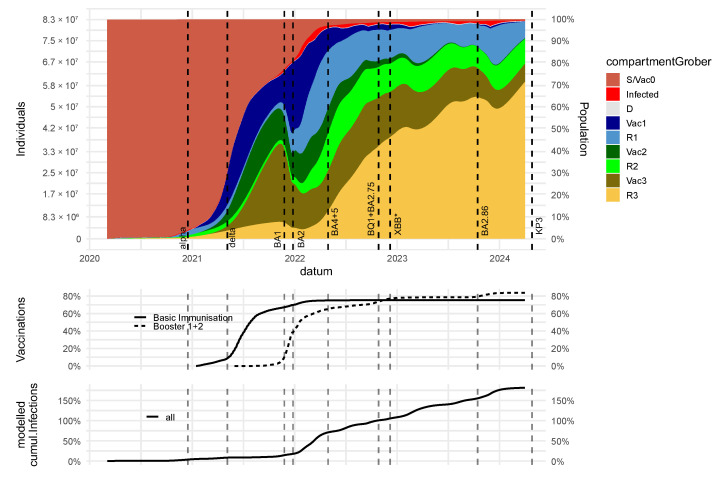
Estimated dynamics of modeled immune states for Germany: We present estimated dynamics of modeled different immune states for Germany (**upper panel**: *S*/*Vac_0_* = immune naïve, *Vac*_3_/*R*_3_ = high risk, *Vac*_2_/*R*_2_ = moderate risk, *Vac*_1_/*R*_1_ = low risk). For comparison, we present the dynamics of the vaccination campaigns (**middle panel**) and the cumulative number of infections (**lower panel**). XBB*: XBB including subvariants.

**Figure 5 viruses-17-00981-f005:**
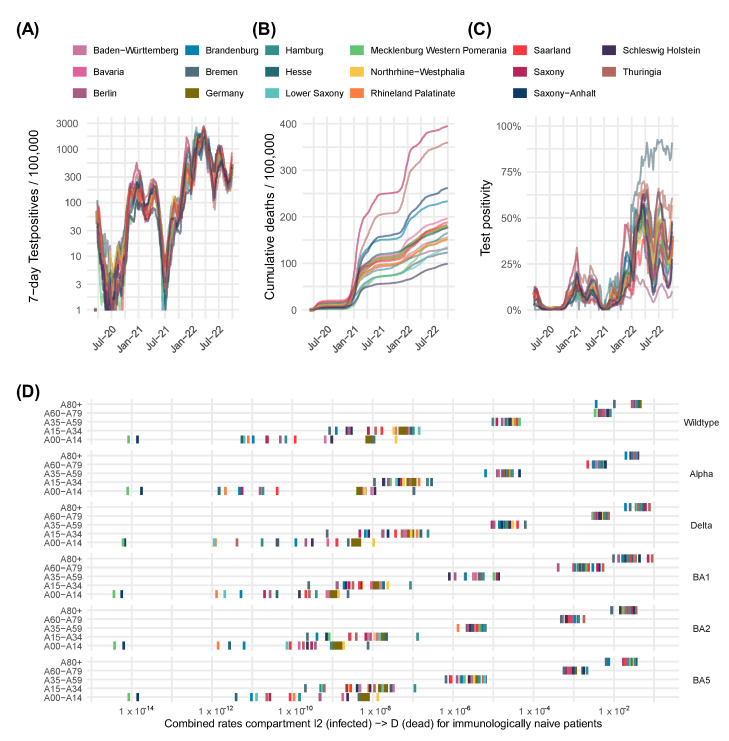
Heterogeneity of the SARS-CoV-2 pandemic between federal states and region-specific parametrization of the model. Considerable between-state differences were observed regarding the course of the pandemic, exemplarily shown by the dynamics of infected subjects (**A**), reported total number of deaths (**B**), and probability of test-positivity reflecting testing policy (**C**). These differences can be explained by state-specific parameters, as exemplarily shown for the transfer rates from the compartment of infected subjects *I*_2_ to the death compartment *D* (**D**). Consistently, federal countries Saxony and Thuringia, having suffered from high excess mortality in the elderly in the alpha and delta waves, show the highest parameter estimates in the oldest age group, 80+. State-specific parameter estimates for hospitalization rates and rates progressing to the ICU are provided in Appendix A, respectively.

**Figure 6 viruses-17-00981-f006:**
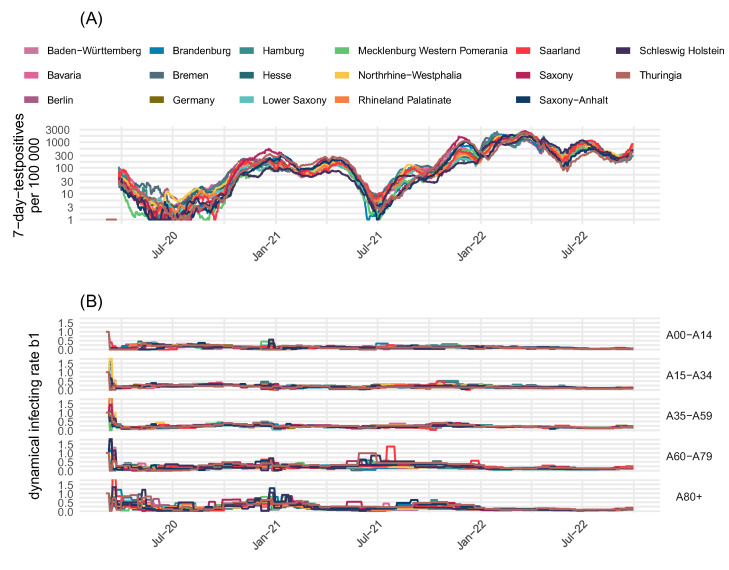
Comparison of between-state heterogeneity of infection dynamics and residual infectivity. (**A**) Reported positive tests per state and (**B**) estimated residual infectivity *b*_1_. Time periods with increased heterogeneity between states are indicated with dashed rectangles. Here, heterogeneity of the residual infectivity is also high.

**Figure 7 viruses-17-00981-f007:**
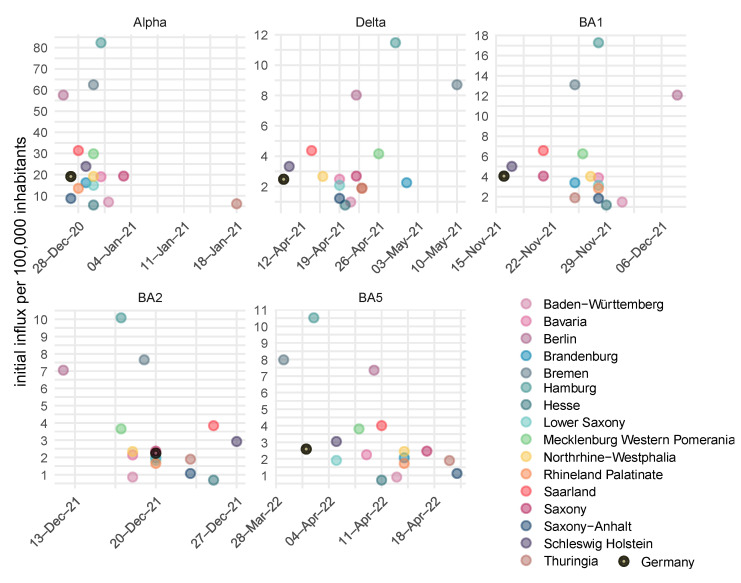
State-specific estimates of the number and time points when new SARS-CoV-2 variants were introduced. For the wildtype, we assumed a constant influx in early 2020.

**Figure 8 viruses-17-00981-f008:**
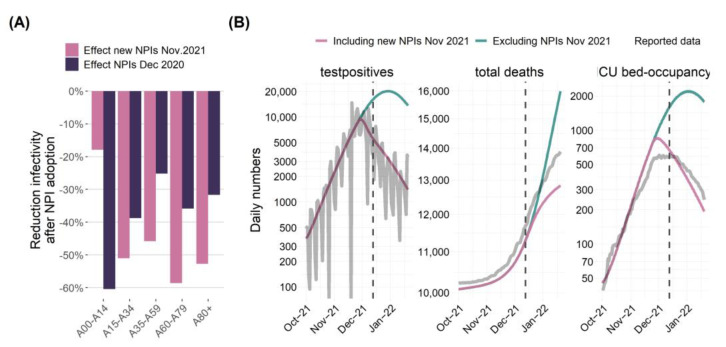
Estimation of the impact of the lockdown measures introduced in November 2021 in Saxony and compared to the December 2020 lockdown. (**A**) Model-based estimate of the decline in infectivity after the introduction of more stringent lockdown measures in December 2020 and November 2021 as a three-week average. While clear and comparable reductions in infectivities were estimated for the age groups older than 14, the reduction for the younger age group was much smaller in 2021 compared to 2020. This is plausible because schools and day care remained open in the 2021 lockdown. (**B**) Modeled scenarios with and without the introduction of lockdown. Without lockdown, the model predicted that infection numbers would increase until the end of December and that the number of deaths would increase in the order of >1000 (https://www.health-atlas.de/documents/34, accessed on 16 June 2025). Model predictions are in reasonable agreement with the actual data. We used data from 4 March 2020 to 13 December 2021 to fit the model (dashed line). NPI: non-pharmaceutical interventions.

**Figure 9 viruses-17-00981-f009:**
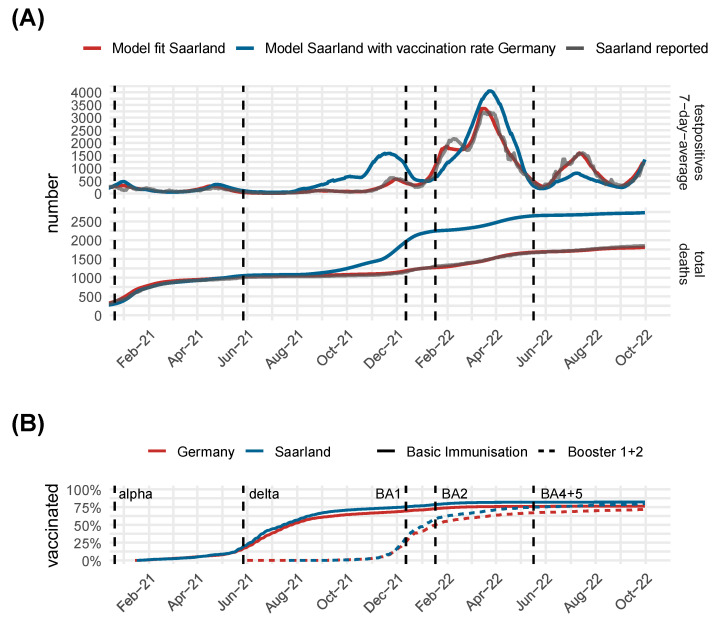
Estimation of the impact of the higher vaccination rate in the federal state of Saarland compared with the German average. (**A**) Comparison of the reported positive tests and COVID-19 deaths (grey) with the model under the observed vaccination rate (red). In blue, we show the hypothetical scenario of the lower average German vaccination rate applied in Saarland. Under this scenario, higher infection numbers of Omicron BA4 + 5 would be expected, but still, the death toll remains lower. (**B**) Cumulative vaccination rates in Saarland compared with Germany. The occurrence of new SARS-CoV-2 variants at frequencies higher than 5% is shown as dashed lines.

**Table 1 viruses-17-00981-t001:** Qualitative properties of the sub-models reflecting different immune statuses. These properties mirror the immune memory induced by vaccination or last infection event.

Compartment	Risk of Infection	Immune Status (see Figure 2)	Risk of Severe Course of Disease (Hospital Ward *H*, ICU Requirement *C*, Death *D*)
*S*, *Vac_0_*	highest	Naive	Highest risk for *H*, *C*, and *D*
*Vac*_1_, *R*_1_	small	Protected	No risk for *C* and *D* and reduced risk for *H* (40% compared to *S*, *Vac_0_*)
*Vac*_2_, *R*_2_	medium		
*Vac*_3_, *R*_3_	high	Waned	Medium risk for *H*, *C*, and *D* (40% compared to *S*, *Vac_0_*)

## Data Availability

Code and data are available via GitHub at https://github.com/GenStatLeipzig/Modelling-dynamics-of-SARS-CoV-2-pandemics-in-Germany and in the Health Atlas at https://www.health-atlas.de/models/40 (accessed on 16 June 2025).

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
