# Peer review of "Modeling the Complete Dynamics of the SARS-CoV-2 Pandemic of Germany and Its Federal States Using Multiple Levels of Data"

_viruses, 2025, doi:10.3390/v17070981_

Round 1
Reviewer 1 Report
Comments and Suggestions for Authors
The authors developed an age-structured model to study the dynamics of SARS-CoV-2 pandemics in Germany and its federal states using multiple levels of data. The authors presented several results. Below are my comments
- I observed several grammatical errors. This must be corrected.
- The GitHub link provided is not the right link. The codebase and the data cannot be found in the link.
- See error statement in lines 229, 237, 383, 375, 348, 327, 314, 316, 386, 393, 427, and 444.
- No keywords in the article.
- Unify the text font.
- Only two literature were cited in the introduction. A proper literature review must be presented because there is a lot of work on the mathematical modeling of COVID-19.
- Using the words cases and infections in your compartment model is confusing. Perhaps cases should be presented as severe.
- The appendix should be supplementary material since it is very lengthy.
- The authors must expand the interpretation of their results. There are a lot of figures without proper biological interpretations.
Author Response
Comment 1: I observed several grammatical errors. This must be corrected.
Authors reply: We thank the reviewer very much for carefully checking our manuscript. We regret the errors. We again checked the manuscript and corrected the detected errors. We would like to mention that some error notifications are related to variables names and abbreviations such as “VirusVariants”.
Changes in manuscript: Errors in Grammar and style improvements were performed throughout the manuscript.
Comment 2: The GitHub link provided is not the right link. The codebase and the data cannot be found in the link.
Authors reply: We are sorry to hear that the intended reviewer access did not work while the GitHub repository was still private. Therefore, we have now made the repository publicly accessible to everyone. We have also reorganized the repository to improve clarity and updated its internet address to https://github.com/GenStatLeipzig/Modelling-dynamics-of-SARS-CoV-2-pandemics-in-Germany.
Changes in manuscript: The website address was updated to https://github.com/GenStatLeipzig/Modelling-dynamics-of-SARS-CoV-2-pandemics-in-Germany
Comment 3: See error statement in lines 229, 237, 383, 375, 348, 327, 314, 316, 386, 393, 427, and 444.
Authors reply: We regret these technical errors. The hyperlinks, which we provided to appendices and figures seem to not work properly in the submitted document.
Changes in manuscript: We corrected the links manually in each of these detected cases as follows:
190 - Appendix C , 195- Appendix G for details ,229 – Appendix D, 237 – Appendix D, 383- Figure 5D, 375 - Figure 5A-C, 348 - Figure 4, 327 - (Figure 3 middle panel), 314- (Figure 3 upper panel), 316 - Figure 3 lower panel, 355- Figure 3D, 386 - Figure 5D, 393 - Figure 7, 427 - Figure 8B, and 444 – Figure 9.
Comment 4: No keywords in the article.
Authors reply: Again, we regret that the keywords were not integrated correctly by the submission system.
Changes in manuscript: We manually added the following keywords:
COVID-19; SARS-CoV-2 epidemiologic models; dark figure; parameter heterogeneity; parametrization; extended multi-compartment SIR-type model; Input-Output Non-Linear Dynamical System; Bayesian knowledge synthesis, Machine-Learning, pandemic preparedness
Comment 5: Unify the text font.
Changes in manuscript: We unified the font.
Comment 6: Only two literatures were cited in the introduction. A proper literature review must be presented because there is a lot of work on the mathematical modeling of COVID-19.
Authors reply: We agree with the reviewer that an improved literature review should be added. Indeed, we found a recent review of SARS-CoV-2 modelling works by Burch et al. and focused on the models presented there. If the reviewer has another specific work in mind, we are happy to add or describe it too.
Changes in manuscript: We added a respective paragraph in the introduction.
“Recently, Burch et al. reviewed mathematical models of COVID-19 vaccination in high-income countries and specifically discussed 47 works [9]. Most of them used deterministic, compartmental models for European countries or North America with a simulated time horizon of 3.5 years or less. Common outcomes included infection numbers, hospital burden and COVID-19 related deaths. Two of the models describe the pandemic situation in Germany [10,11]. Koslow et al. [11] analyzed the relaxation of non-pharmaceutical interventions (NPIs) under the vaccination campaign in Germany for the period June 2021 to March 2022. For the spread of SARS-CoV-2, the authors employed a SIR-type model. Rodiah et al. [10] developed an age-structured deterministic SEIRS model to understand the age- and setting-specific contribution of contacts to transmission during the first 1.5 years of the COVID-19 pandemic in Germany. A pre-pandemic contact matrix has been used there. The model was optimized to fit age-specific SARS-CoV-2 incidences reported by the German National Public Health Institute (Robert Koch Institute).”
We also related our work to those of other models for Germany:
“Here, we chose a full-information approach considering all data in between start of the epidemic 4 March 2020 to 12 September 2024 significantly increasing the modelling period compared to other available models for Germany [10,11].”
Comment 7: Using the words cases and infections in your compartment model is confusing. Perhaps cases should be presented as severe.
Authors reply: The reviewer is right. In our model, “C” corresponds to hospitalized patients, while “I” corresponds to infectious subjects without severe disease conditions. We regret that we did not make this sufficiently clear.
Changes in manuscript: We improved this explanation when introducing the term SECIR hoping that it is sufficiently clear now.
Comment 8: The appendix should be supplementary material since it is very lengthy.
Authors reply: Again, we regret this technical error, which occurred during the submission. It was our intention to provide the supplement as additional material separately from the manuscript.
Changes in manuscript: The respective options were changed during the resubmission.
Comment 9: The authors must expand the interpretation of their results. There are a lot of figures without proper biological interpretations.
Authors reply: We agree that the current presentation of results is a bit sparse.
Changes in manuscript: We extended the descriptions of findings throughout the results section. In particular, we explained all figures in more detail.
Reviewer 2 Report
Comments and Suggestions for Authors
see the attached file

Author Response
Comment 1: Some of these equations need a more detailed explanation, especially that of uninfected susceptible subjects SSM.
Authors reply: We agree to explain this equation in more detail.
Changes in manuscript: We added the following sentence in the appendix before (B1):
“We do not consider births, non-COVID-19 related deaths or transitions between age-compartments due to aging. This is justified by the relative stability of the considered age-groups during the pandemic (see Appendix O). Consequently, there are only effluxes from the susceptible compartments due to (1) new infections (influx), (2) transition from S to E compartment due to infection and (3) additional effluxes due to vaccination (transition to vaccinated compartments)”
Throughout the supplement material, we added further explanations to the equations, where considered useful. We are happy to perform additional improvements if requested by the reviewer.
Comment 2: There is no birth rate there and this is strange since early ages are considered in the model. The factors that influence its evolution appear to be not related to other state variables.
Authors reply: The reviewer is correct that we did not account for birth rates. Similarly, we did not include transitions between age compartments due to aging or effluxes due to deaths. In summary, our model implicitly assumes that the composition of the age groups remains constant. We believe this assumption is justified, as we consider relatively broad age categories whose sizes did not change during the relatively short duration of the pandemic.
Changes in manuscript: At the beginning of Appendix A, we acknowledge that adding aging, births and deaths are not considered. We added Appendix O showing almost constant distributions of age-groups during the course of the pandemic to justify this assumption.
Under limitations (discussion), we acknowledge that this issue could be improved by later versions of the model: “Finally, we so far did not consider births, non-COVID-19 related deaths and aging in our model. This is justified by an almost constant distribution of our age-groups during the course of the pandemic (Appendix O) but could be refined at later versions of the model.”
Round 2
Reviewer 1 Report
Comments and Suggestions for Authors
No further comments for the authors.